# THE EMERGENCE OF THE LEFT-RIGHT ASYMMETRY IN PREDICTING BRAIN ACTIVITY FROM LLMS' REPRESENTATIONS SPECIFICALLY CORRELATES WITH THEIR FORMAL LINGUISTIC COMPETENCE

## ABSTRACT

When humans and large language models (LLMs) process the same text, activations in the LLMs correlate with brain activity measured, e.g., with functional magnetic resonance imaging (fMRI). Moreover, as the training of an LLM progresses, the performance in predicting brain activity from its internal activations improves more in the left hemisphere than in the right one. The aim of the present work is to understand which kind of competence acquired by the LLMs underlies the emergence of this left-right asymmetry. Using the OLMo-2 7B language model at various training checkpoints and fMRI data from English participants, we compare the evolution of the left-right asymmetry in brain scores alongside performance on several benchmarks. We observe that the asymmetry co-emerges with the formal linguistic abilities of the LLM. These abilities are demonstrated in two ways: by the model's capacity to assign a higher probability to an acceptable sentence than to a grammatically unacceptable one within a minimal contrasting pair, or its ability to produce well-formed text. On the opposite, the left-right asymmetry does not correlate with the performance on arithmetic or Dyck language tasks; nor with text-based tasks involving world knowledge and reasoning. We generalize these results to another family of LLMs (Pythia) and another language, namely French. Our observations indicate that the left-right asymmetry in brain predictivity matches the progress in formal linguistic competence (knowledge of linguistic patterns).

## 1 INTRODUCTION

The success of large language models (LLMs) in natural language processing tasks has generated a lot of interest in understanding their internal representations and their alignment with human brain activity. Brain activations, measured with functional magnetic resonance, magnetoencephalography, or electrocorticography, in humans listening to or reading a text can be predicted from the internal activity of LLMs fed with the same text (Jain & Huth, 2018; Toneva & Wehbe, 2019; Schrimpf et al., 2021; Caucheteux & King, 2022; Goldstein et al., 2022; Pasquiou et al., 2023; Antonello et al., 2024). In the encoding model approach, brain activations are regressed on the hidden neural activations from an LLM and the resulting model is used to compute cross-validated correlations at each voxel (brain scores).

Early studies (Huth et al., 2016; Jain & Huth, 2018; Caucheteux et al., 2021; Pasquiou et al., 2023) reported brain score maps that were very symmetrical, with similar brain scores in the right and in the left hemisphere, an odd finding given all the evidence for left hemispheric dominance for language. For instance, in their seminal paper, Huth et al. (2016) noted that "One striking aspect of our atlas is that the distribution of semantically selective areas is relatively symmetrical across the two cerebral hemispheres. This finding is inconsistent with human lesion studies that support the idea that semantic representation is lateralized to the left hemisphere."

Recently, however, Bonnasse-Gahot & Pallier (2024) showed that these symmetrical results, observed with word embeddings or small, first generation LLMs, disappear in larger and more performant models. More precisely, brain score maps exhibited an increasing left-right hemispheric

asymmetry when LLMs increased in number of parameters and in performance on NLP tasks. Furthermore, this left-right asymmetry also emerged for a given LLM alongside its training. The relationship between amount of training and left-right asymmetry showed a phase transition profile that is reminiscent of those that have been observed in LLMs' performance on several benchmarks (Chen et al., 2023).

The present work aims at understanding what competence, acquired during training, drives the emergence of the left-right asymmetry in brain score. We conduct a series of experiments designed to track the evolution of linguistic and non-linguistic capabilities of LLMs as a function of training progression, and we study their relationship to left-right asymmetry in brain score. In our initial experiment, we systematically investigate how the performance of an LLM on a set of carefully constructed benchmarks evolves with training. This set includes two linguistic benchmarks (BLiMP, Warstadt et al., 2020 and Zorro,Huebner et al., 2021) and two non-linguistic benchmarks (specifically, Arithmetic and Dyck language tasks), all designed as minimal pair tasks to isolate specific competencies. Our analyses reveal a striking correlation: as training progresses, the emergence of the left-right dominance in brain score maps closely mirrors the improvement in performance on the linguistic benchmarks, but not on the non-linguistic benchmarks.

In a follow-up experiment, we focus on text-based tasks, contrasting between formal linguistic competence (knowledge of linguistic rules and patterns) and functional linguistic competence (understanding and using language in the world), a distinction proposed by Mahowald et al. (2024). To further assess the model's formal competence, we use it to generate texts at different checkpoints during training, which we feed to another model fine-tuned to evaluate the linguistic acceptability of sentences. To assess functional competence, we test the LLM on conceptual and reasoning benchmarks, namely, ARC (Clark et al., 2018), and Hellaswag (Zellers et al., 2019). The results show that the trajectory of linguistic acceptability correlates with the left-right transition in brain scores, unlike the performance on functional benchmarks.

The results reported above are based on OLMo-2 7B model (OLMo et al., 2024), a recent model for which training checkpoints are available. We show that these results generalize to other models, namely the 2.8b and the 6.9b models from the Pythia family (Biderman et al., 2023). Finally, we generalize to another language, French, and replicate the finding that the left-right asymmetry aligns better with a formal test (grammar) than with a functional one (Hellaswag).

Collectively, these results support the hypothesis that the emergence of the left-right asymmetry in LLMs' brain predictivity is a direct reflection of their formal linguistic abilities.

## 2    MATERIALS AND METHODS

### 2.1    BRAIN IMAGING DATA

The experiments reported in this paper rely on functional magnetic resonance data provided by the multilingual project *Le Petit Prince*, in which English, French and Mandarin Chinese speakers were scanned while listening for a bit more that an hour and a half to an audiobook of The Little Prince (Li et al., 2022). The presentation of the audiobook was split into 9 parts of approximately equal duration, during which functional images of the full brain were acquired every 2 s. Following the procedure of Bonnasse-Gahot & Pallier (2024), after spatially normalizing these images into a common space and resampling them at $4\times4\times4$mm, we average the time-series across participants (high-pass filtered with a cut-off of 128 s and standardized in each voxel), to obtain an average English subject (from 49 participants) and an average French subject (from 28 participants).

### 2.2    LANGUAGE MODELS

The main large language model used in this study is the 7B-parameter version of OLMo-2 (OLMo et al., 2024), released by Allen AI[1]. As far as we know, this is the best open-weight model under 10B parameters that releases a sufficient number of training checkpoints that allow to study the evolution of the performance of an LLM during training. The model has 32 layers and a hidden size of 4096. We consider 10 checkpoints from the base model: the first checkpoint available, after training on

---

[1]OLMo-2-1124-7B is available at `https://huggingface.co/allenai/OLMo-2-1124-7B`

1B tokens; the final checkpoint of their Stage 1 pretraining phase, which is the main part of their pretraining, corresponding to 1 epoch on the OLMo-Mix-1124 dataset (approximately 4T tokens); and 8 intermediate checkpoints log-spaced between these two extremes (using the closest available checkpoint). The main experiments in this study are based on this model. To check that the results are not specific to it, we also run some experiments on Pythia 2.8b and Pythia 6.9b (Biderman et al., 2023).

## 2.3 BRAIN SCORES

For each voxel, we compute a brain score that quantifies how well we can predict the brain activity from the activations of the large language model, using the pipeline made available by Bonnasse-Gahot & Pallier (2024) at `https://github.com/l-bg/llms_brain_lateralization`. In brief, this pipeline follows a standard cross-validation approach where the fMRI time series in a given voxel is fit with a linear model, regularized using ridge regression, on the activations obtained at a given layer of the LLM. The brain score associated with a given voxel is the maximum correlation associated with the best layer. Finally, left and right hemisphere brain scores are obtained by averaging correlations from voxels located in the left and right hemispheres respectively. The main figures of the paper display the left-right asymmetry in brain scores taking into account the 25% most reliable voxels, associated with the highest inter-subject correlation. In Appendix, we also provide supplementary figures based on brain scores including all voxels (whole brain).)

## 2.4 EXPERIMENTS

### EXPERIMENT 1: MINIMAL PAIRS BENCHMARKS

The linguistic minimal pairs benchmarks, BLiMP (Warstadt et al., 2020) and Zorro (Huebner et al., 2021), provide minimal pairs of sentences in which the first sentence, but not the second, is deemed acceptable by English native speakers.

A causal language model has learned to compute, given a string of words as the context, the probability distribution of the next word, over the vocabulary of a language. The log probability of a sentence is then computed as the sum of the log probabilities of each word of the sentence. For a given minimal pair, the most acceptable sentence is the one associated with the highest probability according to the model. For a given task, the overall accuracy is calculated as the proportion (reported as a number between 0 and 1) of times the model correctly assigns a higher probability to the correct or most acceptable sentence.

In order to evaluate the competence of the LLM on non-linguistic tasks, but using the same framework as the BLiMP and Zorro benchmarks, we designed two other benchmarks that likewise involve minimal pairs. An arithmetic benchmark evaluates the ability of a model to assign a higher probability to a correct addition or multiplication, than to an incorrect version. Moreover, a Dyck language benchmark evaluates the model's ability to assign a higher probability to a sequence of well-parenthesized list of parentheses, compared to a version that involves the same elements but with some errors introduced by permuting some neighboring parentheses.

• **BLiMP** (Warstadt et al., 2020) provides 67,000 minimal pairs of English sentences, grouped into 67 paradigms of 1,000 pairs each, isolating specific phenomena in syntax, morphology, or semantics (see Warstadt et al., 2020, Table 4, for examples of each of these phenomena). Here is an example (the asterisk denotes the ungrammatical string), from the *left branch island simple question* dataset:

(1) a. Whose hat should Tonya wear?
   b. * Whose should Tonya wear hat?

• **Zorro** (Huebner et al., 2021) is similar to BLiMP but uses a restricted vocabulary assumed to be known by a 6-year-old English child. Data consist of 22 files containing 4,000 sentences each (*ie* 2,000 minimal pairs). Here is an example from the *agreement subject verb across relative clause* paradigm:

(2) a. The book that I like is poor.

b. * The books that I like is poor.

• The **Arithmetic** benchmark consists of an 'addition' subtask and a 'multiplication' subtask. Each subtask involves 2048 pairs of statements, one correct and one incorrect. The 'addition' task considers statements of the form $x + y = z$, where $x$ and $y$ are randomly chosen between 0 and 1000. In the correct statement, $z$ is indeed the sum of $x$ and $y$, whereas in the incorrect version, an error randomly drawn from the set $[-10, -2, -1, 1, 2, 10]$ is added to the actual sum. In the 'multiplication' task, statements are of the the form $x \times y = z$, where $x$ and $y$ are randomly chosen between 0 and 100. In the correct one, $z$ is the product of $x$ and $y$, whereas in the incorrect one, as for the previous 'addition' task, we add to the product an error randomly drawn from the set $[-10, -2, -1, 1, 2, 10]$. The final accuracy is the mean accuracy over these two addition and multiplication tasks. Here is an example of a minimal pair:

(3)   a.  $36 \times 41 = 1476$
      b. *  $36 \times 41 = 1486$

• The **Dyck** benchmark consists of three sub-benchmarks, based on the Dyck-1, Dyck-2, and Dyck-3 languages, which are formal languages that describe the balanced nesting of opening and closing parentheses (or other types of brackets). Here, Dyck-1 language involves the open and close parentheses '(' and ')', Dyck-2 uses parentheses and square brackets '(', '[', ']' and ')', and Dyck-3 parentheses, square brackets and curly brackets '(', '[', '{', ']', ')' and '}'. For each subtask, we randomly generate 1024 minimal pairs of sentences of length 32. For a given pair, the correct version is well-parenthesized, whereas we introduce errors in the incorrect version by randomly permuting two neighboring elements in the second half of the sentence, so that the two sentences share the same beginning and the same elements overall. Below is an example of such a minimal pair from the Dyck-3 benchmark. The final accuracy is the average of the accuracy on these three subtasks.

(4)   a.  ( ( ) [ ] ) ( ) { [ ] } { } { } { ( ) } ( ) [ ] ( { { } } ) [ ]
      b. *  ( ( ) [ ] ) ( ) { [ ] } { } { } ( ) ( } ( [ ) ] ( { { } } [ ) ]

EXPERIMENT 2: OTHER BENCHMARKS

Next, we further explore the relationship between brain score asymmetry and performance on language tasks with three tests tapping formal aspects of language, namely the linguistic acceptability of text generated by the model, and functional aspects, assessing world knowledge and reasoning using existing evaluation benchmarks (Zellers et al., 2019; Clark et al., 2018).

• **Linguistic acceptability of text generations**: We assess the evolution during training of the linguistic acceptability of the text generated by an LLM (base model). For each checkpoint during training, the LLM is asked to generate a continuation, between 192 and 256 tokens, from one of the following five prompts: 'Why not', 'Are you', 'This is', 'Alice was', 'Bob went'. Texts are generated five times for each prompt, each time with a different initial seed. All generated texts will be available in a repository on the GitHub page of the project. Appendix C provides samples from one trial of one prompt, for all 10 checkpoints. In order to automatically evaluate the acceptability of the generated texts, we use another LLM that was fine-tuned to output the linguistic acceptability of a sentence. This latter model[2] is a version of DeBERTa-v3-large (He et al., 2023) fine-tuned on the CoLA dataset (Warstadt et al., 2019). The Corpus of Linguistic Acceptability (CoLA) is a widely used benchmark dataset for evaluating the ability of natural language processing models to judge the grammatical acceptability of English sentences. It consists of more than 10,000 English sentences labeled as either grammatical or ungrammatical (see Warstadt et al., 2019, Table 3, for samples). The generated text is first split into sentences (using the `sent_tokenize` function from `nltk` Python package, Bird et al., 2009), then each sentence is fed into this fine-tuned LLM. The final linguistic acceptability score of the text is the mean score over all sentences in the text.

• **Hellaswag** (Zellers et al., 2019) is a completion test that assesses commonsense natural language inference. Given an event description, the language model must select the most likely followup

---

[2]Found on the Hugging Face hub, available at `https://huggingface.co/yiiino/deberta-v3-large-cola`

among four choices. The 10,000 sentences were created to be very easy for humans but difficult for NLP systems (that existed around the publication date).

- **ARC** (Clark et al., 2018) provides a question set which contains 7,787 natural, grade-school science questions (authored for human tests), assessing knowledge and reasoning according to the authors. The ARC question set is partitioned into a **Challenge Set** and an **Easy Set**.

EXPERIMENT 3: REPLICATION WITH PYTHIA MODELS

In order to check that the results are not specific to OLMo-2-1124-7B, we replicate experiment 1, evaluating brain scores and performance on the four minimal-pair benchmarks (BLiMP, Zorro, Arithmetics and Dyck) using two other models: Pythia-2.8B and Pythia-6.9B (Biderman et al., 2023). We consider 10 checkpoints during training, equally log-spaced, from step 16 (about 30M tokens) to step 143000 (about 300B tokens, the last step available of the pretraining phase).

EXPERIMENT 4: REPLICATION IN ANOTHER LANGUAGE (FRENCH)

In this experiment, we replicate experiment 1, computing the brain scores of the OLMo-2 model using the French text and the average French fMRI subject. Given that OLMo-2 was mostly trained on English content (OLMo et al., 2024), with occasional texts from other languages, we expect its linguistic competence in French to lag that in English. We assess the formal linguistic competence using the fr-grammar task and the functional competence using the French Hellaswag (both tasks come from the **FrenchBench** benchmark; Faysse et al., 2025). We then compare the trajectories of the performance on these tests during training to the left-right asymmetry in brain predictivity.

## 3 RESULTS

### 3.1 EXPERIMENT 1: MINIMAL PAIRS BENCHMARKS

Fig. 1 shows the evolution during training of the left-right asymmetry in brain scores, computed here with OLMo-2 7B, and the performance of this model on the minimal pairs benchmarks (BLiMP, Zorro, Arithmetic and Dyck). First, we observe a phase transition, that is an abrupt change, for the left-right asymmetry in brain predictivity (blue curve). The brain score in the left hemisphere becomes stronger than in the right as the model is trained on more tokens, reproducing in more details the behavior reported by Bonnasse-Gahot & Pallier (2024) (in their Fig. B10). As for the performance on the four minimal pairs tests, the BLiMP and Zorro benchmarks (left panels) show a phase transition in the same interval as the left-right asymmetry, while the scores on the non-linguistics tests, Arithmetic and Dyck (right panels), do not follow the same pattern. The amount of training where formal linguistic abilities emerge is around 10B tokens, consistent with what Tigges et al. (2024) found with the Pythia family (see also our own results with Pythia below, section 3.4).

We further focus on the sub-tasks of BLiMP labeled "morphology", "syntax", "syntax_semantics", and "semantics" by the authors of this benchmark. The evolution of performance split across these four categories is displayed on Supplementary Fig. B.2. The emergence of left-right asymmetry aligns slightly more closely with the performance on syntactic tests than with those pertaining to morphology or semantics. This suggests a particular salience of syntactic processing in driving the observed brain-LLM alignment.

### 3.2 EXPERIMENT 2: OTHER BENCHMARKS

Fig. 2 shows the results of additional tests which are not based on minimal pairs (ARC, Hellaswag, and Linguistic acceptability). On the one hand, ARC and Hellaswag, the high-level comprehension benchmarks, are not aligned with the left-right brain score asymmetry. On the other hand, the linguistic acceptability of texts generated by the model at various checkpoints exhibits a transition between 5B and 13B tokens that closely matches the left-right asymmetry. Examples of texts generated by the model at successive checkpoints, presented in Appendix C, confirm that it is in this range that the model starts to produce well-formed prose.

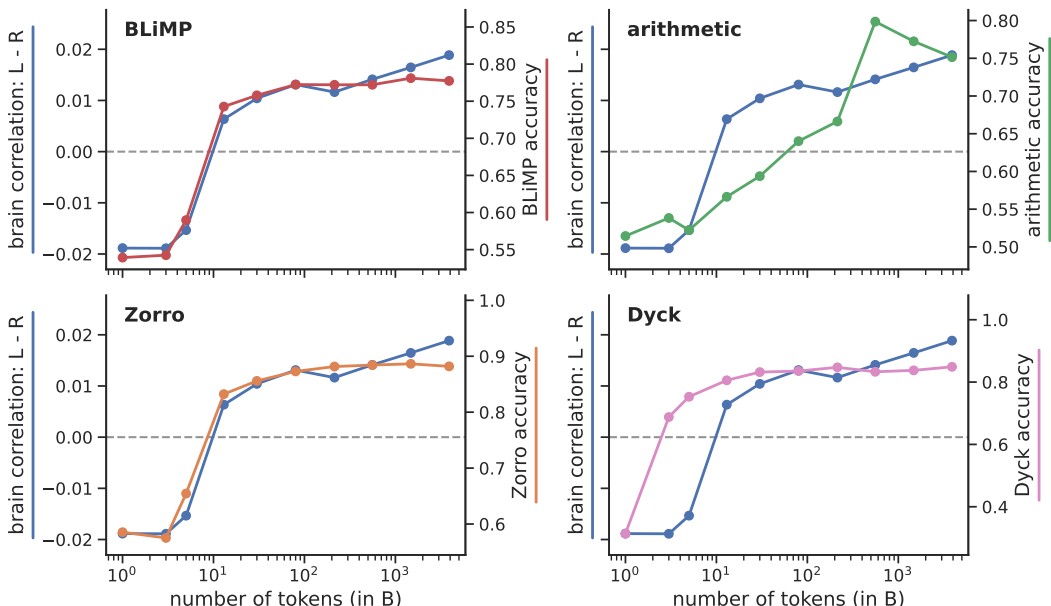

Figure 1: **Phase transitions during training: "minimal pairs" benchmarks.** Each panel displays the left-right hemispheric asymmetry in brain scores (blue curve, repeated across panels) and the performance on a given test, as a function of the number of tokens seen during training (on a log scale). The left panels show the performance on the linguistic tests, BLiMP and Zorro, and the right panels show the performance on the non-linguistic tests, Arithmetic and Dyck. Model used: OLMo-2-1124-7B. Brain scores are computed on the 25% most reliable voxels (see Fig. B.1 for whole brain results). To help compare the transitions, the benchmarks curves were scaled along the y-axis to match the left-right asymmetry curve, by minimizing the mean absolute vertical distance.

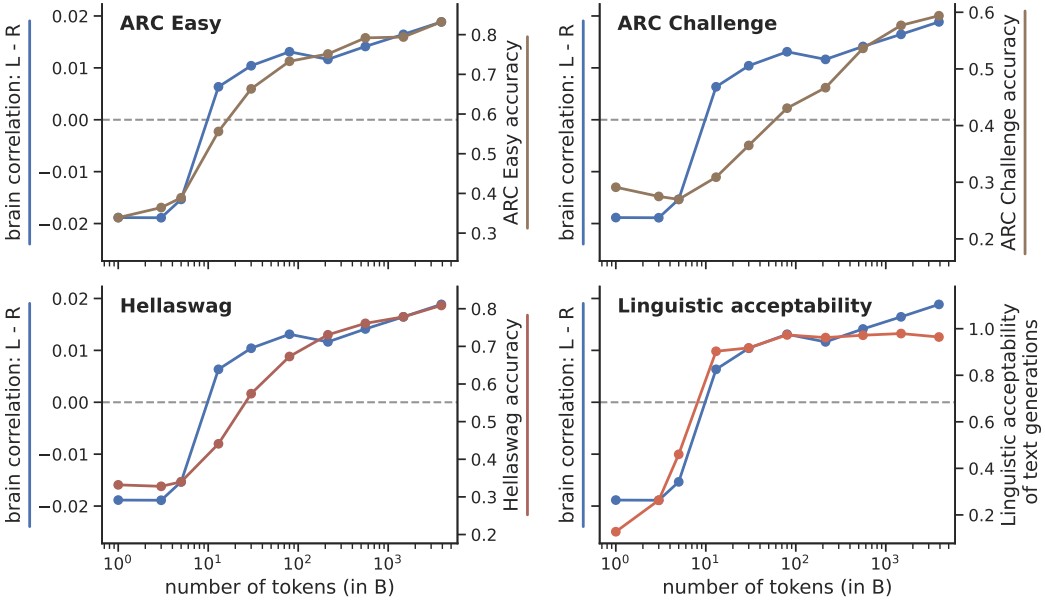

Figure 2: **Left-right hemispheric asymmetry aligns with the acquisition of formal linguistic competence, but not with high-level language comprehension**. Formal competence is assessed by automatically evaluating the linguistic acceptability of text generated at each training checkpoint. The Hellaswag and ARC benchmarks assess world knowledge and commonsense reasoning.

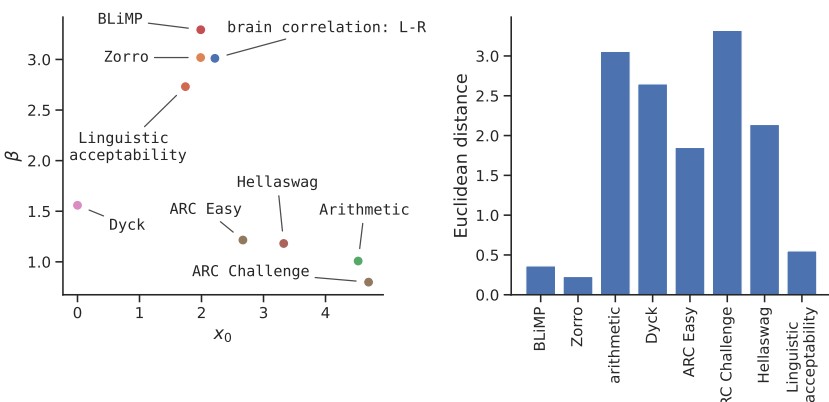

Figure 3: **Quantitative comparison of the evolution of the left-right hemispheric brain scores and the various performance trajectories.** (Left) After fitting a sigmoid to the evolution of a given quantity, we plot the results on a $(x_0, \beta)$ plane, where $x_0$ is the location of the transition along the log(number of tokens) axis, and $\beta$ the slope of the change (Right) Euclidean distance between the location on the $(x_0, \beta)$ plane of each benchmark and left-right asymmetry.

### 3.3 QUANTITATIVE ANALYSIS OF THE ALIGNMENT BETWEEN TRAJECTORIES.

The performance on the various benchmarks increases with the amount of tokens seen during training, as does the left-right asymmetry. On a $x$-log scale, this results in sigmoid shaped curves. To provide a quantitative comparison between all the different trajectories, for each curve displayed on Fig. 1 and 2, we fit a sigmoid in order to locate the phase transition $x_0$ on the $x$-axis (log of the number of tokens) and its slope $\beta$. The fit is obtained by minimizing the mean square error between the target relevant curve and the following sigmoidal function: $y = y_{\min} + (y_{\max} - y_{\min})/(1 + \exp(-\beta(x - x_0)))$, where $x$ is the logarithm of the number of tokens seen during training. Supplementary Fig. B.3 provides a full visualization of these fits.

The location $x_0$ of the transition and its slope $\beta$ can then be used to quantitatively compare all the different transitions. Panel (a) of Fig. 3 shows the location of each benchmark in the $(x_0, \beta)$ space; Panel (b) shows the distance of each benchmark to the parameters of the brain asymmetry transition. This quantitatively confirms that the left-right asymmetry aligns well with the acquisition of formal linguistic competence, but not with high-level language comprehension or other competences such as arithmetic ability.

### 3.4 EXPERIMENT 3: REPLICATIONS WITH PYTHIA MODELS.

To check that the results are not specific to the OLMo-2 7B model, we replicate Experiment 1 with two models from the Pythia family, which also provides checkpoints during training. Fig. 4 shows the results for these models. Again, the left-right asymmetry aligns remarkably well with the acquisition of the formal competence of the model, but not with the functional ones.

### 3.5 EXPERIMENT 4: REPLICATION IN ANOTHER LANGUAGE (FRENCH).

In this last experiment, we check if the previous results are replicable in another language. To this end, we use French data from Le Petit Prince (French text, French fMRI average subject) and evaluate the main model, OLMo-2 7B, on French benchmarks. The results are displayed on Fig. 5.

As expected given that that OLMo-2 7B was primarily trained on English, its development of formal French competence is delayed and progresses more slowly compared to English, and so does the left-right hemispheric asymmetry in brain scores. However, like in English, the evolution of grammatical competence follows more closely the left-right asymmetry than does the performance on French Hellaswag, assessing knowledge and reasoning. Supplementary Fig. B.5 comparing the parameters of fitted sigmoids of all the relevant quantities, confirms this.

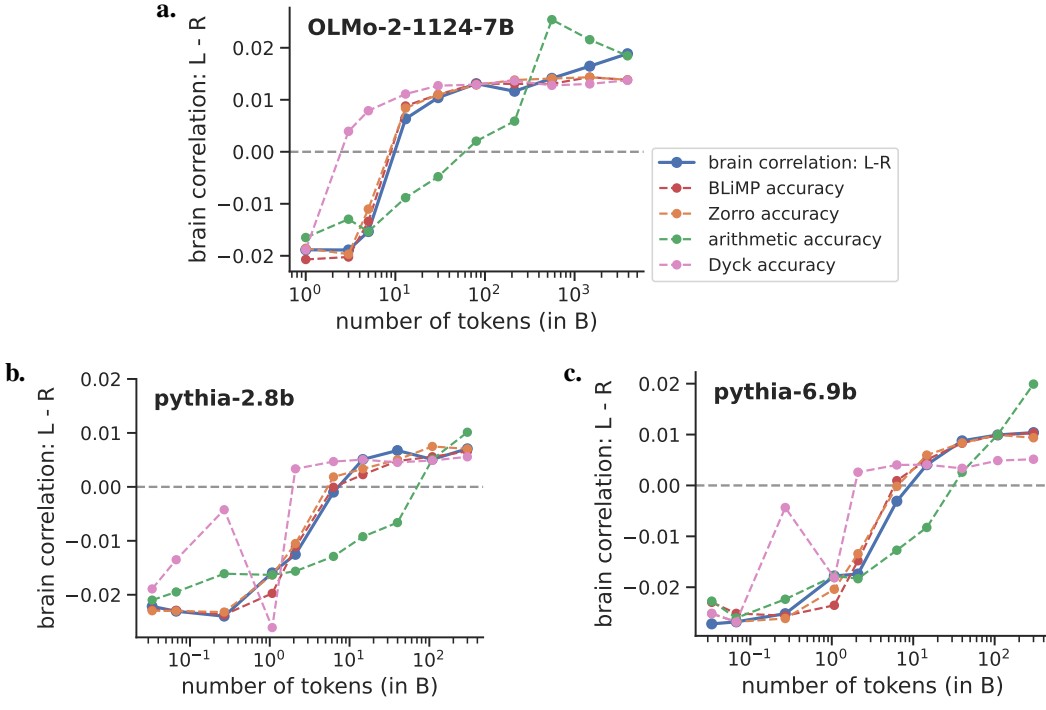

Figure 4: **Generalization to other language models.** Results from minimal pairs benchmarks on OLMo-2-1124-7B extend to Pythia-2.8b & 6.9b models. Panel (a) reproduces data shown in Fig. 1 but with all curves superimposed. Panels (b) and (c) show the results for the Pythia models (figures split by tasks are provided in supplementary Fig. B.4).

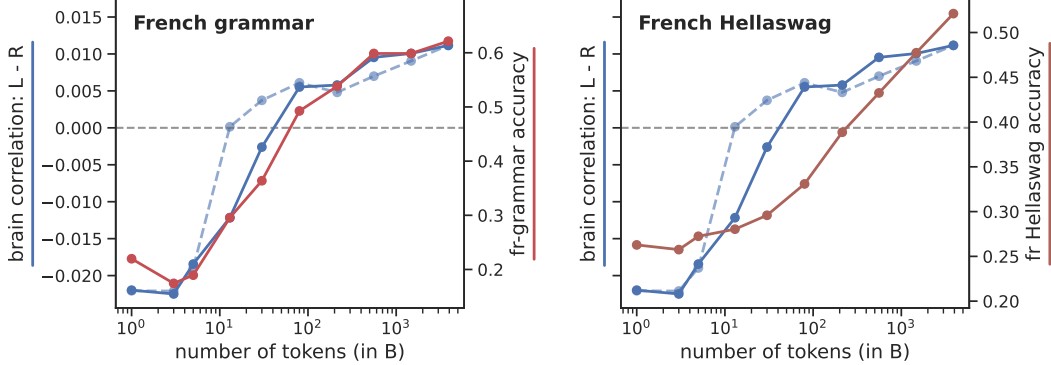

Figure 5: **The evolution of the left-right asymmetry in brain predictivity in French subjects follows the acquisition of formal competence in this language by the LLM.** The model is OLMo-2-1124-7B again. The blue line represents the evolution of the left-right asymmetry computed with on a French average subject. For comparison purposes, the dotted blue line reproduces the left-right asymmetry previously observed in English and displayed in Fig. 1. In the left panel, the red curve tracks performance on the fr-grammar benchmark which measures French formal linguistic competence; in the right panel, the red curve shows the performance on French Hellaswag which assesses functional competence. (Note: both tests are part of the FrenchBench evaluation benchmark (Faysse et al., 2025)). See supplementary Fig. B.6 for results on the whole brain volume.

## 4 DISCUSSION

In order to understand the origin of the emergence of the left-right asymmetry in brain scores with training, we ran a number of benchmarks on LLMs (OLMo-2 7B, Pythia 2.8b and Pythia 6.9b) at different training checkpoints. First, we reproduce and extend Bonnasse-Gahot & Pallier (2024) finding that the left-right asymmetry emerges with training, to new models and with a more fine-grained resolution of training steps. Second, we show that the left-right asymmetry emergence co-occurs with the emergence of formal linguistic abilities in LLMs, attested either by their ability to assign a higher probability to an acceptable sentence than to a grammatically unacceptable one within a minimal contrasting pair (BLiMP and Zorro benchmarks on Fig. 1), or their capacity to produce well-formed text (Fig. 2). Furthermore, the trajectory of the left-right asymmetry with training did not correlate with arithmetic or formal language (Dyck) tasks (Fig. 1), nor with tasks involving world knowledge and reasoning (ARC and Hellaswag; see Fig. 2).

In a recent study, AlKhamissi et al. (2025) compared the developmental trajectories of brain scores, formal linguistic competence and functional competence (Mahowald et al., 2024) and showed three successive phase transitions: brain scores raise first, followed by formal competence, and only later by functional competence. Here, we also find that functional competence is acquired later during training compared to formal competence, but we find that the left-right asymmetry strikingly aligns with the trajectory of formal performance (see Figs. 1, 2, 4, and 5), contrary to the absolute brain score which start to increase before formal competence (see AlKhamissi et al., 2025, Fig. 4).

Among all our tasks, one was especially easy to acquire: the Dyck languages based on nested parentheses. One possibility is that this is due to in-context learning: Olsson et al. (2022) proposed that some attention heads ("induction heads") enable a model to recognize and complete patterns based on previous occurrences in a prompt. They reported that transformer language models undergo a "phase change" early in training, during which induction heads form and simultaneously in-context learning improves dramatically. This mechanism could be at play for the Dyck languages in our experiment. Another possibility could be due to low-level reasons such as bigrams violations in the ungrammatical sequences of parentheses (e.g. { ) in Dyck-3). In any case, the underlying phenomena is acquired early by the LLM, well before the left-right transition.

One point of caution is in order. One should not jump to the conclusion that brains scores are only driven by syntactic knowledge. Kauf et al. (2024), manipulating sentences by altering word order, removing words, or changing semantic content, observed that brain scores were more affected by changing semantic content. This led them to claim that "lexical-semantic content of the sentence (largely carried by content words) rather than the sentence's syntactic form (conveyed via word order or function words) is primarily responsible for the ANN-to-brain similarity". It would be interesting to check how these manipulations impact the left-right asymmetry.

Although we focused in this paper on a global property, the left-right hemispheric asymmetry, the relationship between brain scores and the linguistic performance of models at different training stages should eventually be evaluated more finely at the level of brain regions. This type of approach has been applied very recently to visual processing by Raugel et al. (2025), who observed that brain scores in various regions have different trajectories as a function of the amount of training. More precisely, they reported that the model they study, Dino v3, first aligns with the early representations of the sensory cortices, and needs more training data to align with higher-level regions. It would be worth investigating whether similar links between the brain and LLMs at different training steps exist for language.

Finally, although we show that the left-right asymmetry in brain predictivity displays a strong change during training, the functional competence of the LLM keeps improving after this transition (see scores on ARC and Hellaswag benchmarks on Fig.2). Whether, how, and where this translates into better brain scores is a question for future research.

### REPRODUCIBILITY STATEMENT

fMRI data come from the publicly available fMRI corpus *Le Petit Prince*. All pretrained language models were downloaded from Hugging Face through the *transformers* interface. To assess the performance of OLMo-2 7B on Hellaswag, ARC, and FrenchBench, we relied on EleutherAI's evaluation tools. More detailed information is available in Appendix A. The full code to reproduce the

analyses presented in this paper will be available on GitHub upon acceptance and is now available as a downloadable supplementary material.

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

## A  COMPUTER CODE

All fMRI data come from the publicly available fMRI corpus *Le Petit Prince* (Li et al., 2022)[3]. The Python 3.10 code written for the present project relies on the following libraries: `transformers v4.56.0` (Wolf et al., 2020), `scikit_learn v1.6.1` (Pedregosa et al., 2011), `nilearn v0.11.1` (contributors), `Pytorch v2.7.1` (Paszke et al., 2019), `nltk v3.9.1` (Bird et al., 2009), `matplotlib v3.10.3` (Hunter, 2007), `seaborn v0.13.2` (Waskom, 2021), `numpy v2.0.2` (Van Der Walt et al., 2011), `pandas v2.2.3` (McKinney et al., 2010), `scipy v1.15.2` (Virtanen et al., 2020). All pretrained models were downloaded from Hugging Face through the `transformers` interface. To assess the performance of OLMo-2 7B on Hellaswag, ARC, and FrenchBench, we relie on EleutherAI's evaluation tools `lm_eval v0.4.9` (Gao et al., 2024)[4]. The code will be available on GitHub upon acceptance and is available as a downloadable supplementary material.

## B  SUPPLEMENTARY FIGURES

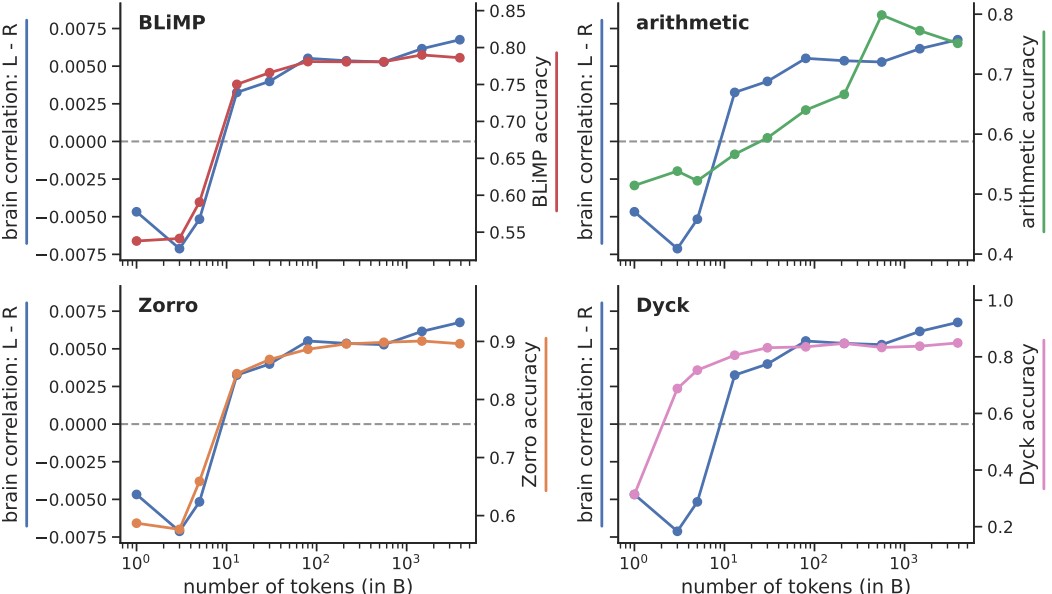

Figure B.1: **Phase transition during training: "minimal pairs" benchmarks.** Same as Fig. 1, but for the whole brain volume.

[3]https://openneuro.org/datasets/ds003643/versions/2.0.5
[4]https://github.com/EleutherAI/lm-evaluation-harness

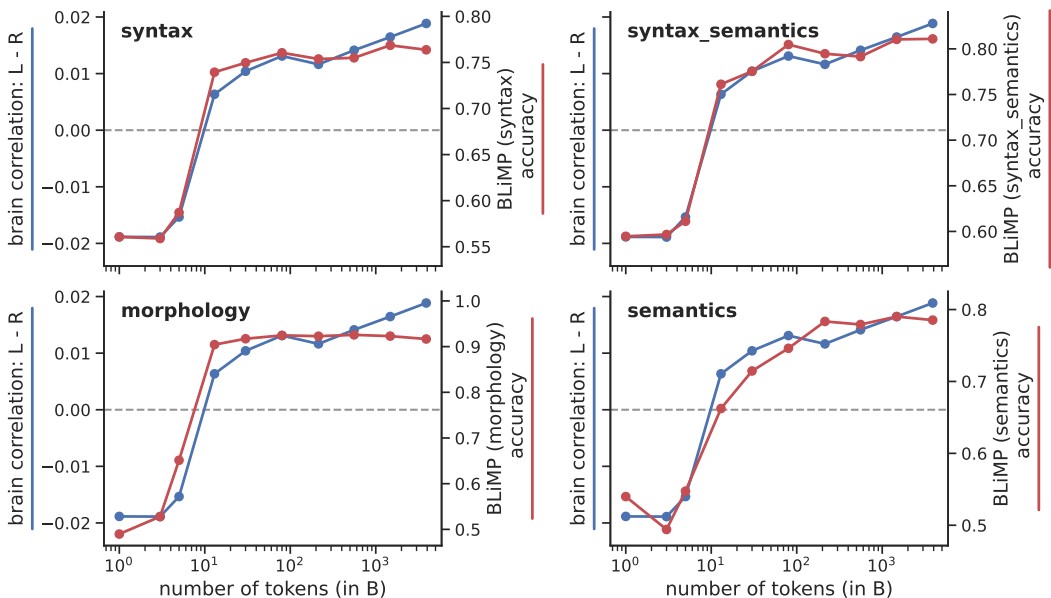

Figure B.2: **Results split by BLiMP subtasks**. BLiMP labels minimals pairs in 4 different fields ("syntax", "syntax_semantic", "morphology", "semantic"). For each field, a panel displays the evolution of performance during training, along with the left-right brain predictivity asymmetry (as in Fig. 1).

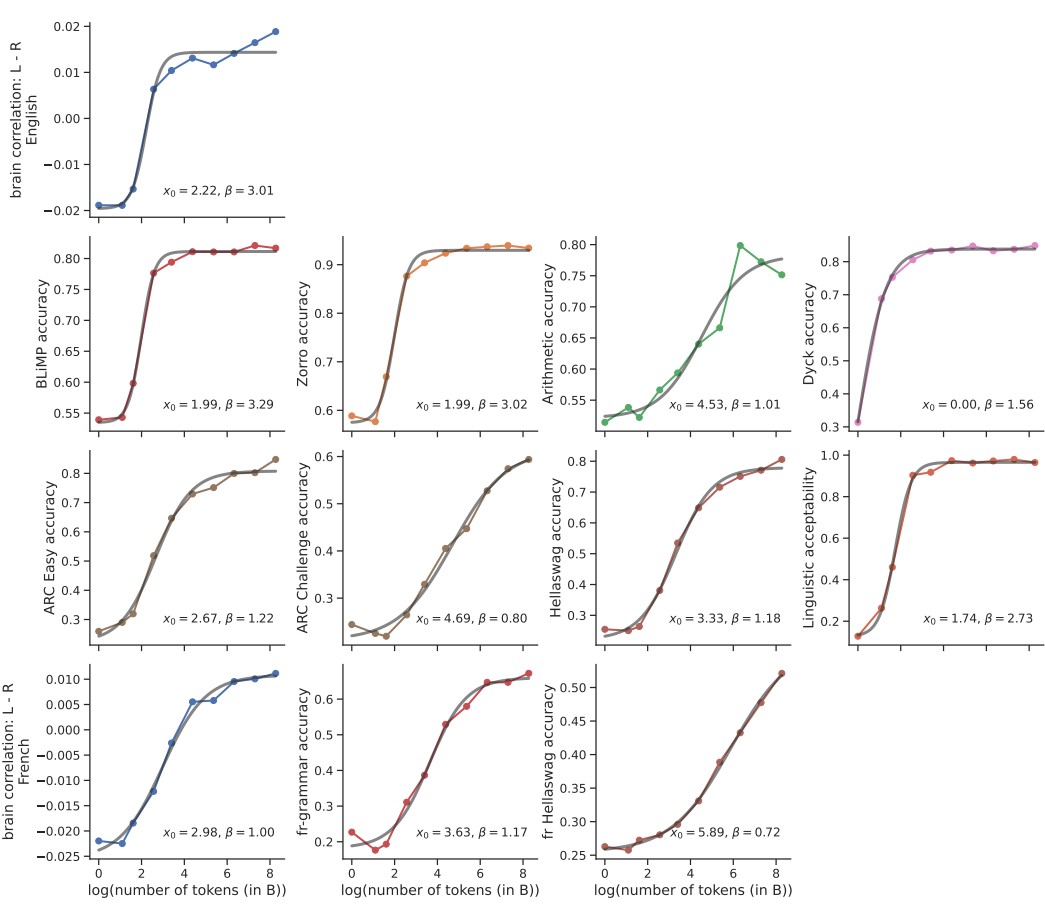

Figure B.3: **Fitting sigmoid to the evolution of various quantities during the training of the OLMo-2 7B language model**. The colored line corresponds to the true values, while the gray line is the resulting fitted sigmoidal curve. See Fig. 3 for a quantitative comparison between the dynamics of the left-right asymmetry in brain predictivity and the performance of the LLM on the various evaluations.

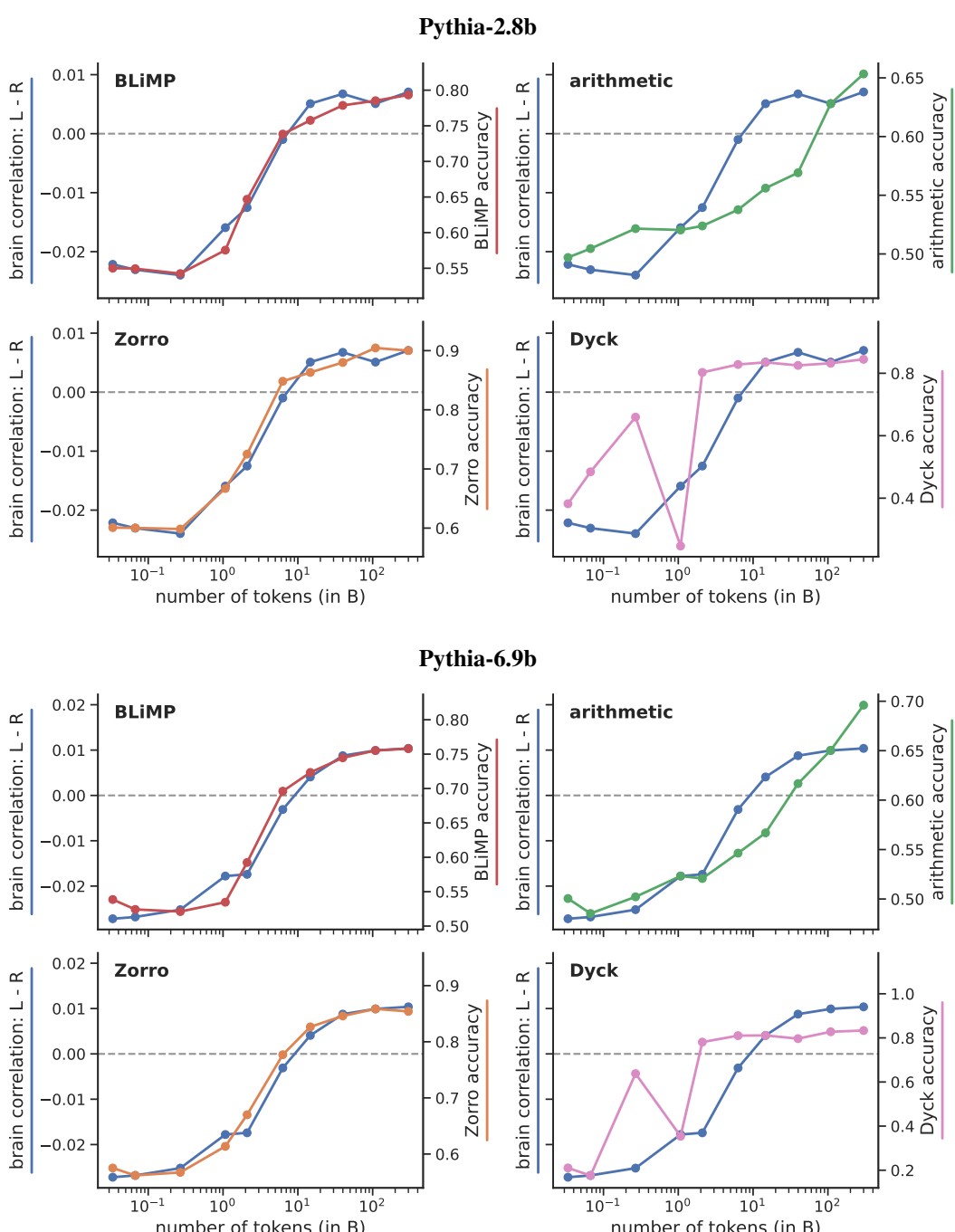

Figure B.4: **Generalization to other language models.** Same as Fig. 1 but for the Pythia-2.8b (top panel) and Pythia-6.9b (bottom panel) models. In each case, the trajectory of left-right brain asymmetry aligns well with the evolution of the performance on linguistic minimal pairs benchmarks, but not on the non-linguistic ones.

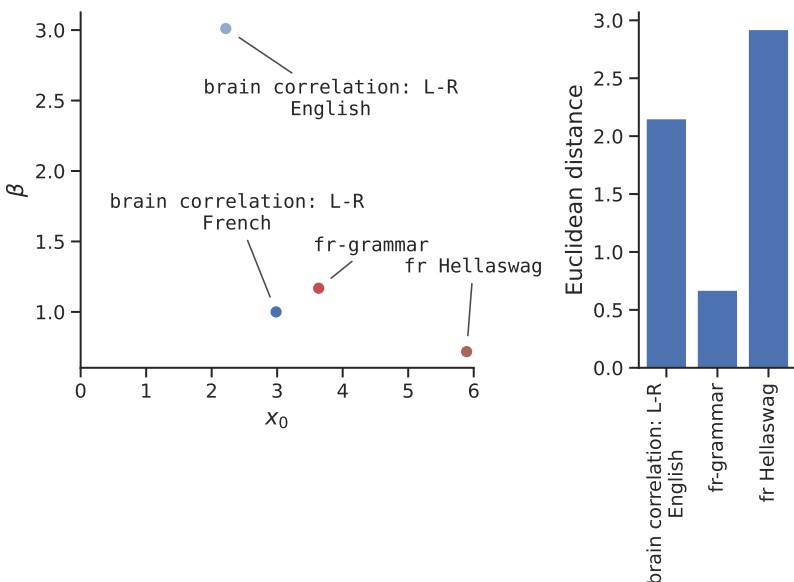

Figure B.5: **Quantitative comparison of the evolution of the left-right hemispheric brain scores in French participants and the various performance trajectories.** Similar analysis as in Fig. 3 but for the French data. Left-right asymmetry in English participants is also provided as a comparison. The right panel shows the Euclidean distance between the location on the $(x_0, \beta)$ plane of each point on the left panel and the left-right asymmetry in French participants. See Fig. B.3, bottom row, for a full visualization of the sigmoid fits.

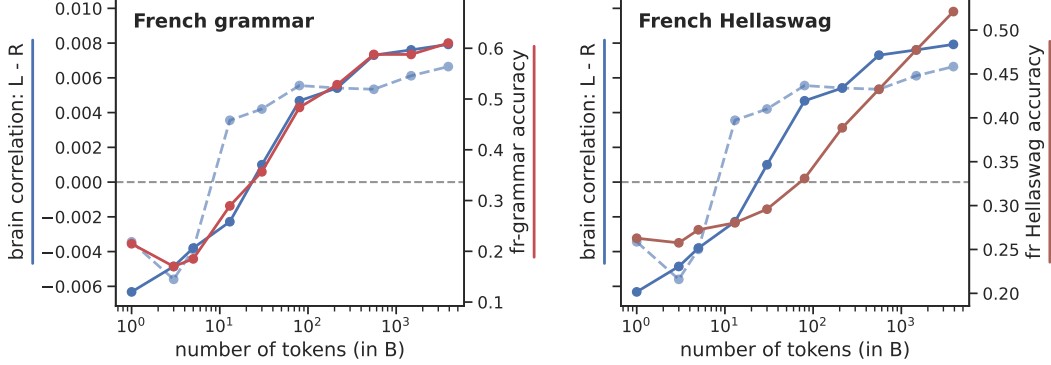

Figure B.6: **The evolution of the left-right asymmetry in brain predictivity in French subjects follows the acquisition of formal competence in this language by the LLM**. Same as Fig. 5, but for the whole brain volume.

## C EXAMPLES OF TEXTS GENERATED BY OLMO-2 7B FOR DIFFERENT TRAINING CHECKPOINTS

This section provides examples of texts generated by the OLMo-2 7B model for the 10 different training checkpoints selected for analysis in the present study. These are samples from one trial (one random seed for the generation) of one prompt (namely, 'Bob went'). All generated texts will be available in a repository on the GitHub page of the project. Each sample starts with a number (1B, 3B, 5B, etc) indicating the number of tokens (in billions) seen by the corresponding checkpoint.

```
---------------------------------------- 1B ----------------------------------------
Bob went:.  , the other for the't the the the new one was ".  When are's it post.  The was
in the said.  2.  The  was the 7 of 2.8, this as " and it is the (  had a " that the two) was
17.  It is that did the this is some, in a.  (1.   and the same of all of some't your with the
people in the as.  The1 is that is at the end, the their.  times of  in the time.  3's all in
the's their of that will was an this is a does , and the in in the 2, 3.1.  And it is in the
the other (.  ( around the was just the two.   (7 and the the are the was in the two.  The
one was not a "2 by this at 00 in the last the same „ and the other " :  3.  "201 line with 3
are

" in the " was   in ?

For the the9, and the the following, and the, is that are the  7, more„ he
---------------------------------------- 3B ----------------------------------------
Bob went team.  , i do what, and what the reason I did was a pretty stupid job, not the
most romantic.  So we didn't think there is an interesting set-up that was called a real
white-ice-ironist-very- and we may use a "pre-essential."  We've been trying to make a bit of
that and have a strong and unrelent-ly-in-pleed, that's not true- and we have some people who
may find.  But we all know that we have to be the best of this because we are too big.  So we
have no particular idea how we should really be able to make it.  We have a way to take in any
form of thought or do we just to go to a post by the other people.  We think that's just a big
deal, because it's not a good movie.

We've got to be a huge fan of "Lay:  The New Yorker" for a series of decades, and we have been
"in the East," for the movie for three years, where we're going to be in the "good" area in
this and we'll look forward to it."

We're still on the page for us, so we'd like to see how you can get it
---------------------------------------- 5B ----------------------------------------
Bob went' they were the other.

Gosh was the one I knew was "the Man" who was the most talked-about about the man who was
in the first grade.  But that was the point, of course, because it was all about "The Ram"
which was really about "The Great Dane", that was his point.  It was that "The Man" was going
to change.  But as you can see, that was all of those things.  Not every thing that happened
before.  That's all for now.  But I can't tell you this because I knew it was just an awful
dream, because I was talking about that.

Gosh was talking about the 9/1 in the second grade.  I think I did just say it was a good
book with some serious reading and I didn't think the author was an actual writer and I didn't
think it was anything but a good one, that I still think it was a good book, that I didn't
want to, and that I needed to be a writer for the world to find it, and then to be a good
publisher for a lot of people in this book.  I guess the author is a wonderful writer, too,
and we just need to find the great ones.

Gosh, I knew
---------------------------------------- 13B ----------------------------------------
Bob went down the bank of a small town in the state of California.  But after he was done,
it was so far to the west of the town in which he had set off that he called a meeting of the
city police officers and said, "Take it out with a boat and float it." Another officer said,
"Take it with the boat and float it." And again after a while a policeman came by and took
the boat and started it and started again till it was gone.  So this is all that is said about
this event.  Now when this was done this is described as being on this date in February, 1961.

GREG: So what does 1961 have anything to do with this, or is it just another story?

SHER: No, this happened in 1961.  It was already an eventful and fun-filled day in Sacramento,
CA. The city of Santa Barbara had decided to get away from Sacramento and go for a vacation
of some sort, and that was all arranged.  But when the police officers arrived at the scene
of this particular case, they told everybody to keep looking.  Because they said it was a very
cold case, and the temperatures had been very cold, they had decided to make a little boat.
And so they
```

------------------------------------------- 30B -------------------------------------------

Bob went down the hallway and left for the elevator.

There was a loud shrieking coming from Bob's office and so far there was no sign of him there. "Bob is in my office! He just went out!" screamed Lucy as she rushed into the elevator with a sullen look on her face and her arms crossed.       "What's going on?" asked Bob from his desk as he sat behind his computer, "What?  I have to go to class, it's after hour and no students will be in here until after the end of the hour!" Bob stated with a sarcastic grin, Lucy looked at her boss with a puzzled look.       "That means that you have class in two minutes!" Bob stated as he got up to grab his robe and slacks.       Lucy walked out of the office and Bob followed her out and told her to keep an eye out for Bob, then when they got to their destination, Lucy went into Bob's office, which was at the back of his office complex. "Bob, I need your help, the elevator isn't working," she started to explain, but Bob started walking back into his office and opened up his desk.       "I have just gotten my license, how are you supposed to do things without

------------------------------------------- 80B -------------------------------------------

Bob went down the hallway and opened the door.  He was standing on the other side of it, not noticing.  The door was closed and the light from the outside hallway went in as he pushed it open.  There was a black cat laying in the middle of the floor, and a white cat was sniffing at him.  Both of them looked like they knew him.

"Did you get any?"  He said to Bob, but Bob didn't answer.  "Well?  Did you?"

 Bob looked at the floor.  "Yes."

 "And?"  said the black cat, stepping out of the doorframe.

 "Bob is a cat!"  Bob thought, in his head.  He had never thought to tell a black cat that he was a cat, and it was just a big mess, because he wouldn't tell the black cat anything.

 The black cat was not angry or surprised by this at all.  It looked at Bob, and then Bob looked at it.  "What did you get?"  it said, then laughed.

Bob looked at the floor and then at the black cat.  It was the first time he had ever said anything like that, so he didn't know what it was supposed to mean,

------------------------------------------- 214B -------------------------------------------

Bob went down the hallway and opened the back door of the storage room.  "No one's here." he said.  "Go in the front."

She opened the door and looked around.  The place was empty.  "But the back door is locked!" she said, and pointed to the door that she had just locked.

Bob looked and didn't notice anything out of place.  He had been a private investigator for years and nothing ever surprised him.  She closed the door and walked to the front door.  "I'll get the phone," Bob said.

She walked into the kitchen.  It was empty.  "Where are everyone?" she asked.  There were several people in the house.  She was sure she saw someone in the living room watching television.

She walked to the living room.  Someone was watching television but it was in black and white.  The set looked old.

He stood and asked her why she was there.  "Someone was watching the television and it looked old," she said.

"Your friend, the one you brought in," he said.

She looked at him strangely and told him that the friend was gone and the television had always been in color.  Then the television switched to color.  He knew

------------------------------------------- 558B -------------------------------------------

Bob went down the hallway and opened the back door of the storage room.  "No one's here."  He was right.  The room was empty.       "How is that?" said a voice just above Bob's head. Bob turned and looked up.  A ghostly white figure stood a few feet away.       "I...I didn't hear you."    "I've been walking around a lot."     Bob swallowed hard and took a step toward the figure.  The figure smiled.    "Have you seen the young woman?" said Bob. "She is on this side.  You must walk with her."    Bob blinked.  He had no idea what the figure meant by that.   "But where is she?"    The figure pointed toward the wall and raised a hand.  Bob looked at it and saw a crack in the wall leading to a dark hallway.  "The tunnel?" said Bob.  The figure nodded.  "But, how...?"   "You must go through the tunnel." "But the others-"    The figure's smile widened and Bob could feel its emptiness.       "You must go through the tunnel, Bob."    Bob looked at the figure again.  Then, he nodded

------------------------------------------- 1477B -------------------------------------------

Bob went down the hallway of a small, cramped flat which was bare of any other furniture other than the bed and a lamp.  In the bedroom the man said, "Now we're going to play a little game. I am going to blindfold you and we are going to play 'guess who it is.'  You have to guess. It's that simple." After tying Bob's eyes and sitting on the bed next to him Bob said, "What

do you call this game?" "The game goes like this," he said.  "I am going to caress your body
with my hands, you have to identify where I touch you.  If you guess right, I'll caress you a
little longer.  If you guess wrong I'll just stop.  It's like you said, simple.  Ready?" "Wait
just a minute.  I said not if I didn't want to." "You've already lost," he said.

He stood and stripped to his underwear, revealing the same hairy chest, and his cock was
obviously hardening.  He stood next to the bed, pulled off his shorts and stood in front
of Bob, his hard penis in front of Bob's face.  "Let the game begin." He started on Bob's
shoulder, running his finger down Bob's arm,

---------------------------------------- 3896B ----------------------------------------

Bob went down the hallway and opened the back door of the restaurant.  Then he walked out
to his car and saw there was no parking available in the front lot.        "How is that?" he
said to himself.  "They didn't have as many cars when I was there."      Bob pulled into the
front lot and parked near the edge of the parking lot.  He walked into the building and a
woman at the front door said to him, "Good morning."      "Good morning," Bob said.        Bob
went past the front desk and to the restaurant.  As Bob passed the desk, the hostess pointed
out a booth.        "I've got a table," Bob thought.  He walked to the booth and sat down.  He
ordered his meal and sat back and relaxed.  He had a good book to read and was reading an
article about an upcoming movie based on the book by the famous author.  A waiter took Bob's
order and asked him if he wanted another drink.        "No, thanks.  I've got a coke here," Bob
replied.        The waiter took his glass and poured in some ice and then put in some coke from
a glass bottle.  Bob took a sip and said, "I don't think it needs any more coke."      "Sorry,

