# OpenReview forum: "The emergence of the left-right asymmetry in predicting brain activity from LLMs' representations specifically correlates with their formal linguistic competence"
_ICLR.cc/2026/Conference — Submitted to ICLR 2026_

### Official Review · Reviewer_K8zy · 2025-10-23

**Soundness:** 3
**Presentation:** 4
**Contribution:** 2
**Rating:** 2
**Confidence:** 5

**Summary:**

This paper investigates why left-right hemispheric asymmetry arises when large language model (LLM) activations are used to predict human brain activity during language processing. Using training checkpoints from the OLMo-2 and Pythia model families, the authors track how brain predictivity evolves alongside model competence. They find that left-hemisphere dominance co-emerges with the model's acquisition of formal linguistic skills (e.g., syntax, grammatical acceptability) rather than with non-linguistic or reasoning abilities. The authors clain that this relationship generalizes across models, languages (English and French), and datasets. The results suggest that formal linguistic competence, not world knowledge or reasoning, drives LLM–brain alignment asymmetry.

**Strengths:**

1. The study offers a cognitive-level explanation for hemispheric asymmetry in brain–LLM alignment, bridging neuroscience and computational linguistics.
2. Tracking asymmetry evolution through training provides rare temporal insight into how representational properties emerge in LLMs.
3. The separation of formal vs. functional linguistic competence provides a theoretically grounded interpretation of model–brain correspondence.

**Weaknesses:**

1. Authors chose two language acceptability tasks (BLiMP and Zorro). Expts should have been done with more linguistic tasks to claim "formal competence (knowledge of linguistic patterns)" in the abstract. The same holds for functional competence tasks -- just two tasks do not justify a broad claim. For French the comparison is just on one linguistic and one non-linguistic task. Aren't ARC and Hellaswag linguistic tasks where the claim does not seem to hold -- what is the definition of a linguistic task?
2. The study depends on one fMRI dataset where participants were listening to an audiobook. Will these observations and insights hold on other fMRI datasets where participants do other tasks in unclear?
3. Line 168 says "ARC and Hellaswag, the high-level comprehension benchmarks, are not aligned with the left-right brain score asymmetry." From Fig 2, to me it looks like alignment is high for ARC easy task. Is there a metric and a threshold for this alignment based on which these claims of alignment vs not are made?
4. Given Fig 4, Fig 1 is redundant.
5. French expt should have been done with some model trained specifically for French. French results do not seem to be convincing.
6. It would be nice to do this experimentation to observe patterns across different brain regions rather than just hemisphere level results.

**Questions:**

1. It would be nice to know x_0 vs \beta fit for the 2 french expts.
2. You argue that left-right asymmetry reflects formal linguistic competence rather than functional competence. How do you rule out the possibility that this asymmetry instead reflects differences in training data distribution or token-level statistics rather than linguistic structure
3. You report correlations between training progression and asymmetry. Did you test lag effects (e.g., whether changes in linguistic competence precede or follow changes in brain predictivity)?
4. Do you think these results would hold for languages with different lateralization patterns (e.g., logographic or morphologically rich languages)?

---

> ### Author Response · Authors · 2025-11-24
>
> Thank you for your review. Here is a point-by-point response to your comments in the Weaknesses section and your questions.
>
> > Authors chose two language acceptability tasks (BLiMP and Zorro). Expts should have been done with more linguistic tasks to claim "formal competence (knowledge of linguistic patterns)" in the abstract. The same holds for functional competence tasks -- just two tasks do not justify a broad claim. For French the comparison is just on one linguistic and one non-linguistic task. Aren't ARC and Hellaswag linguistic tasks where the claim does not seem to hold -- what is the definition of a linguistic task?
>
> We believe there was some confusion due to our description of the various tasks in the abstract and the introduction, that we tried to tackle in the revised version of the manuscript. ARC and Hellaswag are text-based tasks that involves higher level understanding (reasoning or world-knowledge). The point is exactly this: the left-right asymmetry in brain scores aligns remarkably well with the acquisition of formal competence, as assessed by three independent benchmarks, namely BLiMP (which is itself composed of 67 individual datasets that isolate specific phenomenon in syntax, morphology, or semantics), Zorro (encompassing 13 paradigms) and by evaluating the linguistic acceptability of texts generated by various checkpoints during training; in contrast, the left-right asymmetry does not align with other competences, assessed by four tasks, two non-linguistic benchmarks (Dyck and arithmetic) as well as two linguistic-mediated benchmark ARC and Hellaswag that requires higher level understanding (reasoning or world-knowledge). We also extend this to French, with two tasks, one that corresponds to formal aspects, and another one to more functional aspects.
>
> We have updated the abstract and the introduction to make this clearer.
>
> > The study depends on one fMRI dataset where participants were listening to an audiobook. Will these observations and insights hold on other fMRI datasets where participants do other tasks in unclear?
>
> Here we are studying naturalistic language comprehension in two independent datasets, one in English and one in French. There is little reason to expect that the results are specific to these datasets and would not extend to other fMRI dataset of passive natural language understanding. Our focus here is on language understanding, and we do not have any claim nor expectation for other tasks, but this might be interesting to consider in the future.
>
> > Line 168 says "ARC and Hellaswag, the high-level comprehension benchmarks, are not aligned with the left-right brain score asymmetry." From Fig 2, to me it looks like alignment is high for ARC easy task. Is there a metric and a threshold for this alignment based on which these claims of alignment vs not are made?
>
> Yes, that is precisely the goal and results of the analysis that uses sigmoid fits to quantitatively compare the different trajectories (see Fig. 3 and new Section 3.3 in the revised version), which quantitatively confirms that ARC easy is indeed closer than the ARC challenge, but it is still far compared to the three formal competence measures (accuracies on BLiMP and Zorro, and the linguistic acceptability scores). It is true that there is no threshold here, but we see that the points relating to formal competence nicely cluster around the left-right asymmetry in brain scores in Fig. 3, contrary to all the other quantities.
>
> > Given Fig 4, Fig 1 is redundant.
>
> Although this is partially true, partially because the full figure, Fig. 1, gives the proper y-axis for each benchmark, we believe that Fig. 1 is also easier to read. The main aim of Fig. 4 is to make it easy to compare between the models in the replication experiment.
>
> > French expt should have been done with some model trained specifically for French. French results do not seem to be convincing.
>
> We believe that this experiment is interesting precisely because the model is not trained specifically on French, so that we expected the formal competence in this language to be delayed compared to the one in English, as happened. We also observe that the left-right asymmetry in French subjects aligns with this formal competence rather than functional competence in French. This rules out a possible global change in the representations that would explain all the alignments if they were to happen at the same time. Empirically, the French results are a nice replication of the English ones. The alignment is particularly striking on the whole brain figure, Fig. B.6 (B.5 in the original submission) provided in Appendix.

---

> ### Author Response · Authors · 2025-11-24
>
> > It would be nice to do this experimentation to observe patterns across different brain regions rather than just hemisphere level results.
>
> We agree this is an interesting avenue for future work. This is actually acknowledged in the Discussion (in  the penultimate paragraph). Note that maps showing the asymmetry at the voxel levels are presented by Bonnasse-Gahot & Pallier (2024) and already provide some information.
>
> > It would be nice to know x_0 vs \\beta fit for the 2 french expts.
>
> Following your comment, we have added  the sigmoid fits for French data to Fig. B.3 (bottom row): the left-right brain asymmetry in French subjects and the performance on the two French benchmarks, fr-grammar and fr Hellaswag. We also provide in Appendix B a new figure (now Fig. B.5) displaying the values of $x_0$ and $\\beta$ on a plane for different relevant quantities as well as all the corresponding Euclidean distances in the $(x_0, \\beta)$ plane to the French left-right asymmetry location on this plane.
>
> This analysis confirms that the performance on formal French benchmark is close to the left-right asymmetry transition of the French fMRI data, compared to functional competence on French as evaluated by the performance on the fr Hellaswag benchmark.
>
> > You argue that left-right asymmetry reflects formal linguistic competence rather than functional competence. How do you rule out the possibility that this asymmetry instead reflects differences in training data distribution or token-level statistics rather than linguistic structure
>
> Do you mean that the text-based functional tests might be more difficult than the formal ones because the training data contains more tokens that overlap with the formal benchmarks? The main difference between BLiMP and Zorro essentially resides in the vocabulary: Zorro relies on highly frequent words that young children are supposed to know. Remarkably, yet, the transitions for BLiMP and Zorro are strikingly aligned (see Fig. 4).
>
> > You report correlations between training progression and asymmetry. Did you test lag effects (e.g., whether changes in linguistic competence precede or follow changes in brain predictivity)?
>
> We don't report correlations between training progression and asymmetry. To estimate the location $x_0$ of the various phase transitions, we fit sigmoids, so that, on Fig. 3, the difference in $x_0$ corresponds to the lag between phenomena. Note though that $x_0$ alone is not enough, as the same $x_0$ but different slope $\\beta$ leads to different qualities of alignment, hence our choice to look at both values.
>
> > Do you think these results would hold for languages with different lateralization patterns (e.g., logographic or morphologically rich languages)?
>
> We are not aware of any language that yields different lateralization patterns in spoken language comprehension. For instance, Malik-Moraleda et al. (2022), across 45 languages and 12 families, show that left-lateralization for language processing is quite universal. Can you provide the relevant references?

---

### Official Review · Reviewer_66aJ · 2025-10-31

**Soundness:** 2
**Presentation:** 2
**Contribution:** 2
**Rating:** 2
**Confidence:** 3

**Summary:**

This paper investigates the origins and potential causes of the “left-right asymmetry” observed in LM-brain alignment --- i.e., the phenomenon that LM activations are more aligned with brain activity in humans’ left hemispheres than right hemispheres. The authors investigate how this asymmetry evolves across checkpoints of an LM’s training process, evaluating LMs on a variety of linguistic and non-linguistic tasks, in both English and French. The authors find that the left-right asymmetry co-emerges with the LM’s functional linguistic abilities.

**Strengths:**

The topic of understanding representational alignment across LMs and human brains is interesting and timely.
It is also a plus that the findings are validated in languages beyond English.

**Weaknesses:**

The focus on distinguishing formal/functional competence could have been motivated more clearly. The question “Is the left-right asymmetry driven more by one type of competence than the other?” (l. 066) comes a bit out of nowhere. If we did find that the asymmetry was driven more by one particular kind of competence, what implications would that have? What would be the theoretical importance of investigating this question, either for neuroscience or AI? Since this question is central to the paper’s experiments and analyses, I found the overall high-level takeaways a bit unclear.

Also, acceptability combines many factors, including formal as well as functional language competence. More generally, it would have been useful to cover more tasks beyond syntax for analyzing functional language competence.

I was also unsure about some of the experimental design choices. For example, in the text generation experiment, how were the five seed prompts chosen? What temperature did you use to sample outputs from the models? Did you perform any manual validation of the generated sentences?

Finally, I felt that the focus on tracking left-right asymmetry across training time was not clearly motivated. Couldn’t you also evaluate a large set of models and see how asymmetric brain scores correlate with task performance? To be clear, I think the training time analyses are interesting, but I found it a bit unclear what theoretical question they are answering.

**Questions:**

Below are some questions and more minor suggestions.

It was unclear in the text which of the evaluation datasets were novel, and which were taken from previously published work (e.g., BLiMP, Zorro, HellaSwag, ARC). Please add in-text citations for the datasets that are not novel.

In Section 2.1, does it make sense to average across voxels of different individuals? Does the spatial normalization process take care of this?

Section 2.3 was a bit hard to follow, especially since it is so far ahead of the figures.

It’s not accurate to say that the models are making “acceptability judgments on sentences” for the BLiMP and Zorro benchmarks (l. 440). You are comparing the probabilities of strings, not asking models for acceptability judgments, correct?

---

> ### Author Response · Authors · 2025-11-24
>
> Thank you for your review. Here is a point-by-point response to all your comments and questions.
>
> > The focus on dstinguishing formal/functional competence could have been motivated more clearly. The question “Is the left-right asymmetry driven more by one type of competence than the other?” (l. 066) comes a bit out of nowhere. If we did find that the asymmetry was driven more by one particular kind of competence, what implications would that have? What would be the theoretical importance of investigating this question, either for neuroscience or AI? Since this question is central to the paper’s experiments and analyses, I found the overall high-level takeaways a bit unclear.
>
> The central question of the paper is indeed what kind of competence underlies the emergence of left-right asymmetry. We start with tests opposing language (BLiMP & Zorro ) to non-language (Arithmetics & Dyck), then focus on tests (Arc & Hellaswag) using language but involving higher level  processes (reasoning, world knowledge). Following the terminology of Mahowald et al., we labeled this "functional knowledge", and oppose it to well formedness of strings ("formal knowledge").
>
> As discussed with another reviewer, there is a huge amount of empirical evidence (from aphasics, electrical stimulation, fMRI language studies, ...) that, in the majority of speakers, the left hemisphere is dominant for language processing.  Assuming that the LLMs represent the same type of information (syntactic & semantic & general knowledge) as humans,  it was surprising to some researchers, that early work on brain alignment between LLMs and brain activity (Huth et al., 2016; Jain & Huth, 2018; Caucheteux et al., 2021; Pasquiou et al., 2023) revealed very  symmetrical patterns. Now, after Bonnasse-Gahot & Pallier (2024) we know that larger models recover the asymmetry, and this is also true within a model during its training; the present paper investigates why this is the case.
>
> We apologize that the abstract and introduction did not make this clear enough. We have updated them in the new version to better motivate the work.
>
> > Also, acceptability combines many factors, including formal as well as functional language competence. More generally, it would have been useful to cover more tasks beyond syntax for analyzing functional language competence.
>
> We are aware of the non-trivial relationship between grammaticality and acceptability, but we follow the terminology used by the researchers who designed the benchmarks, and use the formal vs. functional distinction proposed by Mahowald et al. (2024), where functional competence essentially means reasoning and world knowledge.
>
> As for the second part of your comment, do you mean formal language competence? The formal competence is assessed in three different ways that are not strictly  limited to syntax: performance on BLiMP, Zorro, and automatic evaluation of the linguistic acceptability of generated texts at various checkpoints during training. Note that BLiMP and Zorro cover several aspects of linguistic competence, with different classes of tests labeled  "morphology", "syntax", "syntax_semantic" and "semantics", which are analyzed in the Appendix. The results presented on Fig.B.2 suggest that the fit is tighter for the "syntax" and "syntax_semantics" tests. We presented this in appendix as we wanted to remain cautious and not make very strong claims.  Actually, a paragraph in the discussion (lines 460-465) begins with "One point of caution is in order. One should not jump to the conclusion that brains scores are only driven by syntactic knowledge". More work is certainly needed to probe more finely the different types of linguistic knowledge acquired by the model. The current work is a first step in this direction.
>
> > I was also unsure about some of the experimental design choices. For example, in the text generation experiment, how were the five seed prompts chosen? What temperature did you use to sample outputs from the models? Did you perform any manual validation of the generated sentences?
>
> The random seed is set to 12345 before generating the texts. We use the model.generate function from huggingface transformers library, setting the parameter num_return_sequence equal to the number of trials considered in the paper, namely n_trials = 5, with temperature set to the default one (equal to 1.0).
>
> Regarding manual validation, we looked at many texts produced by the model at various checkpoints and the automatic acceptability scores matched our intuition.  We included examples  from one prompt and one trial (seed) in Appendix C for the reader to make his/her own opinion. All generated texts are also provided in the supplementary material of the original submission, and will be available on the GitHub page of the project.

---

> ### Author Response · Authors · 2025-11-24
>
> > Finally, I felt that the focus on tracking left-right asymmetry across training time was not clearly motivated. Couldn’t you also evaluate a large set of models and see how asymmetric brain scores correlate with task performance? To be clear, I think the training time analyses are interesting, but I found it a bit unclear what theoretical question they are answering.
>
> Comparisons between models was the main topic of  the NeurIPS paper by  Bonnasse-Gahot & Pallier (2024). The authors compared many models of various sizes and precisely found a scaling law in the left to right difference between hemispheres as a function of model performance, reconciling modern results using neural network models of brain activity with classic results in neurolinguistics.
>
> Here we explore the origin of this left-right asymmetry by looking at a same model across different checkpoints during training, by comparing its emergence with the performance on several benchmarks. We find that it emerges with the acquisition of formal competence by the model, with a striking quantitative alignment between the two transitions.
>
> As mentioned above, we have modified the manuscript to try and better motivate our work.
>
> > It was unclear in the text which of the evaluation datasets were novel, and which were taken from previously published work (e.g., BLiMP, Zorro, HellaSwag, ARC). Please add in-text citations for the datasets that are not novel.
>
> Following your suggestion, we now provide citations next to each existing benchmark's detailed description.
>
> > In Section 2.1, does it make sense to average across voxels of different individuals? Does the spatial normalization process take care of this?
>
> Spatial normalization is indeed applied before averaging the different individuals (see l. 90--92 of the original submission). Averaging across individuals is quite common in fMRI data analysis (Poldrack, Mumford, & Nichols, 2011, Handbook of functional MRI data analysis).
>
> > Section 2.3 was a bit hard to follow, especially since it is so far ahead of the figures.
>
> Thanks for this remark. We agree and decided to merge this section with the relevant Results section on the quantitative analysis of the alignment between trajectories (new Sec. 3.3), and this indeed improves the readability of the paper.
>
> > It’s not accurate to say that the models are making “acceptability judgments on sentences” for the BLiMP and Zorro benchmarks (l. 440). You are comparing the probabilities of strings, not asking models for acceptability judgments, correct?
>
> You are correct. We are not explicitly asking the model to perform the meta linguistic task of acceptability judgement, but use the probabilities assigned by the model to the strings, as explained in details in the Methods section.  Following your remark, we have modified the abstract and discussion to clarify this point and remove the ambiguity.

---

### Official Review · Reviewer_uPnb · 2025-10-31

**Soundness:** 4
**Presentation:** 3
**Contribution:** 3
**Rating:** 6
**Confidence:** 4

**Summary:**

The paper tests the brain alignment to left and right brain hemispheres (from the Le Petit Prince fMRI dataset) of LLMs over training (OLMo-2 7B, Pythia 2.8 and 6.9B).
This is inspired by a recent finding that LLMs predict the left hemisphere slightly better (Bonnasse-Gahot & Pallier, 2024).
Differences in models' brain alignment scores across hemispheres are compared to their formal and functional competencies via 7 different benchmarks. The authors claim a strong correspondence during training of brain alignment to formal, but not functional competencies.

**Strengths:**

1. correspondence between performance (on BLiMP, Zorro, ARC Easy) seems to mirror the left-right brain alignment asymmetry very closely. I have rarely seen task scores and biology measurements mirror each other this closely. The findings are consistent with a recent study by AlKhamissi et al. (2025), although the correspondence of formal competencies with the left-right asymmetry seems to be even stronger than with brain alignment overall.

2. findings are generalized across multiple models.

3. findings are generalized across multiple languages (English, French).

Code available in supplement, and GitHub release promised upon acceptance.

**Weaknesses:**

My main concern is that the difference in L vs R hemisphere alignment is rather marginal: The overall difference in brain alignment between left and right hemispheres reaches a maximum of 0.02 -- is this a difference that we really think is crucial to investigate? I am genuinely asking, it just seems like a small quantitative phenomena to me but the correspondence with formal competencies is so striking.

From another perspective there is a lack of explanation for why the field should care about this left-right asymmetry. Classic neuroscience studies have claimed a left lateralization of the human language network (e.g., Fedorenko et al.) but with LLMs going well beyond core language processing, it's not obvious to me that we should expect them to be more predictive of the left hemisphere.

It would also have been great to test this on more than one fMRI dataset, but I understand that more models and more datasets would always be nice :)

**Questions:**

Please help us understand why the left-right asymmetry in models' brain alignment is important.

---

> ### Author Response · Authors · 2025-11-24
>
> Thank you for your review. Please find below a point-by-point response to your comments and questions.
>
> > My main concern is that the difference in L vs R hemisphere alignment is rather marginal: The overall difference in brain alignment between left and right hemispheres reaches a maximum of 0.02 -- is this a difference that we really think is crucial to investigate? I am genuinely asking, it just seems like a small quantitative phenomena to me but the correspondence with formal competencies is so striking.
>
> This is an important point to consider.  The effect sizes on brain scores are indeed  modest, but need to be contextualized. First, note that contrary to many authors, we present raw correlations and do not normalize them. Second, the baseline correlation points can be very easy to obtain, as exemplified by the scores obtained by either untrained network or random baselines (see e.g. Schrimpf et al., 2021; Pasquiou et al., 2022; Bonnasse-Gahot & Pallier, 2024; AlKhamissi et al., 2024). The subsequent increases in brain scores are more difficult to obtain. Going from a model with about 100M of parameters to a model with about 10B yields an increase in brain score of 0.06, or 15% in Bonnasse-Gahot & Pallier (2024, Fig. 2), which is similar to what is reported in Antonello et al. (2023, Fig. 1). See also AlKhamissi et al. (2024, Table 2). This does not imply that the linguistic representations are not much better, just that they yield a small increase in brain score. The exact reason is an interesting open question, and might be due to the limitations of fMRI signal which, for example, is very temporally smooth.
>
> > From another perspective there is a lack of explanation for why the field should care about this left-right asymmetry. Classic neuroscience studies have claimed a left lateralization of the human language network (e.g., Fedorenko et al.) but with LLMs going well beyond core language processing, it's not obvious to me that we should expect them to be more predictive of the left hemisphere.
>
> > Please help us understand why the left-right asymmetry in models' brain alignment is important.
>
> There is a huge amount of empirical evidence (from aphasics, electrical stimulation, fMRI language studies, ...) that, in the majority of speakers, the left hemisphere is dominant for language processing.  Assuming that the LLMs represent the same type of information (syntactic & semantic & general knowledge) as humans,  it was surprising to some researchers, that early work on brain alignment between LLMs and brain activity (Huth et al., 2016; Jain & Huth, 2018; Caucheteux et al., 2021; Pasquiou et al., 2023) revealed very  symmetrical patterns. For instance, Huth et al. (Nature, 2016) wrote "One striking aspect of our atlas is that the distribution of semantically selective areas is relatively symmetrical across the two cerebral hemispheres. This finding is inconsistent with human lesion studies that support the idea that semantic representation is lateralized to the left hemisphere."
>
> When using more recent LLMs, Bonnasse-Gahot & Pallier (2024) made an empirical observation: as the NLP performance of the LLMs increased, brain scores become asymmetric, that is, stronger on the left, reconciling this research with classic neurolinguistics.  But their work did not explain why. The present paper is an attempt to understand which competence(s) is acquired by the LLM that explains the emergence of the asymmetry.
>
> This work is only a step towards understanding what drives the alignment between brains and LLMs, (see also AlKhamissi et al., 2025).  For example, Bonnasse-Gahot & Pallier (2024) found that each hemisphere is characterized by its own scaling law, with better models explaining better the right hemisphere too, for reasons that still need to be fully investigated, but which relates to your point that  "LLMs are going well beyond core language processing".
>
> > It would also have been great to test this on more than one fMRI dataset, but I understand that more models and more datasets would always be nice 🙂
>
> Indeed, evidence from more datasets is always nice. Note that here we used two disjoint datasets, the French LPP and the English LPP datasets (different languages, different brain recordings, different participants), and the code we provide can be adapted to other datasets.

---

> > ### Comment · Reviewer_uPnb · 2025-11-25
> >
> > Thank you for your response.
> >
> > I'm still a bit confused as to the intuition: what exactly changes in the LLMs when increasing in scale, that you think leads to this asymmetry?

---

> > > ### Author Response · Authors · 2025-11-25
> > >
> > > What drives the increasing asymmetry with scale across models is not studied here. Our work focuses on a given LLM, and we look at changes during its training. This allows to control for the amount of training data, and avoids confounds such as different architectural choices, different hidden sizes, different training training dataset, etc, making the results simpler to interpret. We see that for a given LLM, the asymmetry is driven by changes in formal linguistic competence. The emergence of the left-right asymmetry with training shows that the LLM represents text in a way that correlates better with representations processed in the left than in the right hemisphere. Let us imagine, for the sake of the argument, that after some training, the LLM suddenly becomes able to represent the syntactic structures of sentences. If those structures are represented in the left hemisphere in humans, then one expects stronger brain scores in the left than in the right. Note also that we only look at the behavior of the LLM and do not dissect its internal working, which would be an interesting future work.

---

### Official Review · Reviewer_V7qQ · 2025-11-01

**Soundness:** 2
**Presentation:** 2
**Contribution:** 2
**Rating:** 4
**Confidence:** 2

**Summary:**

This paper studies why large language models (LLMs) show a left-right hemispheric asymmetry in predicting human brain activity. The main hypothesis is that this asymmetry arises when the LLM gains "formal linguistic competence" (like grammar), not "functional competence" (like reasoning or world knowledge).
The authors analyze how the alignment between LLM representations and fMRI signals changes during training, using the OLMo-2 7B model and both English and French data.

**Strengths:**

1. Clear focus on a specific neuroscience question.
2. Striking correlation between formal linguistic gains and L-R asymmetry.

**Weaknesses:**

1. The automatic scoring of generated text depends on another LLM (DeBERTa), raising concerns about model-induced bias.
2. The quantitative analysis uses very few data points, possibly making the conclusions unstable.
3. Averaging fMRI signals across subjects may mask individual differences.

**Questions:**

1. How do the authors rule out bias in using a model (DeBERTa) to judge another model’s outputs?
2. Why is there no L-R effect for Dyck despite it being a purely formal task?

---

> ### Author Response · Authors · 2025-11-24
>
> Thank you for your review. Please find below our answers to the points you raised.
>
> > How do the authors rule out bias in using a model (DeBERTa) to judge another model’s outputs?
>
> DeBERTa is fine-tuned on CoLA, a corpus on English linguistic acceptability annotated by humans. Although we cannot completely rule out the existence of a bias in the DeBERTa model to judge the output of the generated texts, it is remarkable that the phase transition of the linguistic acceptability of the generated texts is well aligned with the results from the independent BLiMP  and Zorro benchmarks. Finally, we have provided all the generated texts as a supplementary material (which will also be included in the GitHub repository of the project).  In Appendix C, we provided samples from one seed and one prompt, for each of the 10 checkpoints used during training, so that the reader can assess the linguistic acceptability of the outputs. Our own intuitions about the linguistic acceptability of the generated texts match the scores given by the fine-tuned DeBERTa model (see the transition from 5B to 13B tokens).
>
> > The quantitative analysis uses very few data points, possibly making the conclusions unstable.
>
> We consider 10 data points during training. It is not clear that having more datapoints would be very informative for the comparisons where the left-right asymmetry and the benchmarks are clearly misaligned. In the case of the well-aligned curves, such as the left-right brain asymmetry and BLiMP, the 10 data points used during training well encompass the transition in the left-right asymmetry as well as all the other curves from the benchmarks. It is true that in this region, the alignment could be a bit less striking if a point in the middle of the transition would be high in one case and low in another. Following your comment, we have checked this by running all the analyses on one extra point, at 10B tokens during training, right in the transition region for the OLMo-7B model. This point is found to be lying on the segments between 5B and 13B tokens in all figures, and does not impact the analyses with the sigmoid fits shown in Fig. 3. We decided not to add this datapoint on the figures of the revised version as it does not alter the conclusions and it would introduce non-regular sampling on the x-axis which is a bit awkward.
>
> Moreover, our findings are replicated with other models from a different family, with two models from Pythia, where the phase transition is more finely sampled than with OLMo (see Fig. B.4), which gives us faith in the stability of the conclusions.
>
> > Averaging fMRI signals across subjects may mask individual differences.
>
> Bonnasse-Gahot & Pallier (2024) showed that when correlating LLMs with brain activity, results obtained on a few individual participants are in line with the results from the average subject (computed from the 49 participants in the English LPP dataset), the latter being associated with much stronger brain scores. Individual differences in cerebral lateralization for language have been studied extensively for decades and are not the focus of the current paper: such analyses could yet be relevant if the dataset contained an independent assessment of the individual brain lateralization for language, which is not the case.
>
> > Why is there no L-R effect for Dyck despite it being a purely formal task?
>
> The "L-R effect" is computed as the difference in brain scores from the left and right hemispheres and as such is independent of the benchmark (on. Fig. 1, the L-R effect, displayed in blue, is the same across panels). Fig.1 shows that the LLM performance on Dyck language increases earlier than the L-R effect, unlike the performance on BLiMP and Zorro which is strikingly aligned with the L-R transition.

---

### Author Response · Authors · 2025-12-02
**General response**

Dear Area Chair,

We thank all four reviewers again for their valuable feedback. It is frustrating that the discussion period with reviewers ended prematurely and did not allow for more interactions, as we have invested significant effort into the rebuttal.

We took the reviewers' comments into consideration. We improved the abstract and the introduction to better motivate the research question and enhance the presentation of the results. The paper should now read better overall. We have also checked the stability of our results by running the analyses on one additional point right in the transition of the left-right brain asymmetry, which confirms the remarkable alignment with the acquisition of formal linguistic competence that we observe. We have also conducted a new analysis, provided in Appendix, that extends the sigmoid fits that we made for the English experiments to the French ones, and this quantitatively confirms our previous results.

We believe these changes might have led to positive reevaluations of the manuscript. As noted by one reviewer, it is rare to find such a striking alignment between a biological measurement and task scores, here left-right asymmetry in brain scores and performance on formal linguistic tasks. Moreover, our work goes beyond previous works that have correlated LLM and brain activations: it provides some explanations as to why these correlations arise.

---

### Meta-Review · Area_Chair_pqqT · 2026-01-04

**Summary:**

This paper investigates the origins of the left-right hemispheric asymmetry observed when LLM activations are used to predict human brain activity during language processing. Using training checkpoints from OLMo-2 7B and Pythia models, alongside fMRI data from English and French participants listening to audiobooks (Le Petit Prince datasets), the authors track how the asymmetry in brain predictivity evolves during LLM training. The key finding is that left-hemisphere dominance in brain alignment co-emerges specifically with the model's acquisition of formal linguistic competence (syntax, grammaticality) as measured by BLiMP, Zorro, and generated text acceptability—but not with arithmetic, Dyck language, or reasoning/world knowledge benchmarks (ARC, HellaSwag).

The four reviewers had divergent opinions. Reviewer uPnb (Score: 6) found the correspondence between L-R asymmetry and formal competence "striking," noting they had "rarely seen task scores and biology measurements mirror each other this closely." Reviewers 66aJ and K8zy (both Score: 2) raised concerns about motivation, limited task coverage, and experimental design. Reviewer V7qQ (Score: 4) had moderate concerns about methodology but low confidence in their assessment.

**Rationale for decision:** While this paper presents an interesting finding—the striking alignment between left-right hemispheric asymmetry in brain predictivity and the acquisition of formal linguistic competence during LLM training—the overall reviewer consensus does not support acceptance. The key factors informing this decision are:

**1. Split Reviewer Opinion with Strong Skepticism:** Only one of four reviewers (uPnb) was clearly supportive. Two reviewers (66aJ, K8zy) gave scores of 2, and Reviewer K8zy expressed high confidence (5/5) in their negative assessment. The average score (3.5) falls below the acceptance threshold.

**2. Limited Scope of Evidence for Broad Claims:** The paper makes general claims about "formal linguistic competence" but tests only two benchmarks (BLiMP, Zorro) plus generated text acceptability. Similarly, "functional competence" is assessed with only two benchmarks (ARC, HellaSwag). For a paper claiming to identify *which* competence drives brain alignment asymmetry, broader task coverage would strengthen the conclusions.

**3. Presentation and Motivation Issues:** Multiple reviewers found the motivation for distinguishing formal vs. functional competence unclear (66aJ: "comes a bit out of nowhere"). The theoretical implications for neuroscience or AI were not sufficiently developed.

**4. Effect Size Concerns:** While Reviewer uPnb acknowledged the striking correlation, they also questioned whether the 0.02 L-R difference "is crucial to investigate." The authors provided context, but the practical significance of such small effects remains uncertain.

The finding itself is novel and the correlation is indeed striking, as Reviewer uPnb noted. However, the limited scope of evidence relative to the breadth of claims, combined with the skeptical reception from half the reviewers, suggests the work would benefit from additional experiments before publication at a top venue.

**Reviewer Concerns:**

### Addressed by Rebuttal:
- **Motivation and Theoretical Importance** (66aJ, uPnb): Authors clarified the historical context—early LLM-brain alignment work (Huth et al., 2016) found surprisingly symmetric patterns, contradicting classic neurolinguistics on left-hemisphere language dominance. Bonnasse-Gahot & Pallier (2024) showed larger models recover asymmetry; this paper investigates *why* by tracking training dynamics.
- **DeBERTa Bias for Acceptability Scoring** (V7qQ): Authors noted that DeBERTa is fine-tuned on CoLA (human-annotated acceptability), and the phase transition aligns with independent BLiMP/Zorro benchmarks. Generated texts are provided in supplementary material for manual inspection.
- **Few Data Points** (V7qQ): Authors verified stability by adding an extra checkpoint (10B tokens) in the transition region, confirming results. Pythia experiments provide finer sampling of the phase transition (Fig. B.4).
- **Effect Size Significance** (uPnb): Authors contextualized that raw correlations are presented (not normalized), and going from 100M to 10B parameters yields only ~0.06 increase in brain score—the L-R asymmetry of 0.02 is meaningful in this context.
- **ARC Easy Alignment** (K8zy): Authors clarified that sigmoid fit analysis (Fig. 3) quantitatively confirms ARC Easy is still far from formal competence measures, even if visually closer than ARC Challenge.
- **French Model Choice** (K8zy): Authors argued that using a non-French-specific model is informative—it shows formal competence in French is delayed compared to English, and the L-R asymmetry in French subjects aligns with French formal competence, ruling out global representation changes.
- **Training Time vs. Model Comparison** (66aJ): Authors noted that model comparison was the focus of Bonnasse-Gahot & Pallier (2024); this work complements it by controlling for architecture and training data.
- **Sigmoid Fit for French** (K8zy): Authors added sigmoid fits for French data in revised appendix (Fig. B.3, B.5), quantitatively confirming formal French competence aligns with French L-R asymmetry.

### Outstanding Concerns:
- **Limited Task Coverage** (K8zy, 66aJ): Only two formal linguistic benchmarks (BLiMP, Zorro) plus generated text acceptability are used to claim "formal competence." Only two functional competence benchmarks (ARC, HellaSwag) are tested. Reviewers requested broader task coverage for such general claims.
- **Single fMRI Paradigm** (K8zy): The study relies on one paradigm (listening to audiobook). Whether results generalize to other language tasks (reading, production) is unclear.
- **Definition Ambiguity** (66aJ, K8zy): The formal vs. functional competence distinction (from Mahowald et al., 2024) could be motivated more clearly. Acceptability combines multiple factors beyond pure syntax.
- **Brain Region Analysis** (K8zy): Hemisphere-level analysis may miss important regional patterns; voxel-level analysis would strengthen conclusions.
- **Averaging Across Subjects** (V7qQ): Individual differences in lateralization may be masked by averaging, though authors note this is standard practice.

**Reviewer Scores:**

| Reviewer | Initial Score | Predicted Post-Discussion Score |
|----------|---------------|--------------------------------|
| **V7qQ** | 4 (Marginally Below) | **5** - Low confidence (2/5) in assessment. Concerns about DeBERTa bias and data points were reasonably addressed. Stated "would not mind if paper is accepted." |
| **uPnb** | 6 (Marginally Above) | **6-7** - Already positive, finding the correspondence "striking." Engaged in follow-up discussion about intuition. Minor concern about effect size was contextualized by authors. |
| **66aJ** | 2 (Reject) | **3-4** - First-time reviewer with moderate confidence. Concerns about motivation and presentation were addressed with manuscript revisions, but fundamental concerns about task coverage and theoretical importance may persist. |
| **K8zy** | 2 (Reject) | **2-3** - High confidence (5/5) reviewer with strong concerns about limited task coverage, single fMRI dataset, and unconvincing French results. Authors provided additional French sigmoid fits but core concerns about scope remain. Unlikely to change significantly. |

---

### Decision · Program_Chairs · 2026-01-26

Reject